# Energy Resolution from a Silicon Detector’s Interstrip Regions

**DOI:** 10.3390/s24082622

**Published:** 2024-04-19

**Authors:** J. A. Dueñas, A. Cobo, F. Galtarossa, A. Goasduff, D. Mengoni, A. M. Sánchez-Benítez

**Affiliations:** 1Departamento de Ingeniería Eléctrica y Centro de Estudios Avanzados en Física, Matemáticas y Computación, Universidad de Huelva, 21007 Huelva, Spain; 2Grand Accélérateur National d’Ions Lourds (GANIL), Boulevard Henri Becquerel, F-14076 Caen, France; alex.cobo@ganil.fr; 3INFN, Sezione di Padova, 35131 Padova, Italy; franco.galtarossa@lnl.infn.it (F.G.); daniele.mengoni@unipd.it (D.M.); 4INFN Laboratori Nazionali di Legnaro, 35131 Padova, Italy; alain.goasduff@lnl.infn.it; 5Departamento de Ciencias Integradas y Centro de Estudios Avanzados en Física, Matemáticas y Computación, Universidad de Huelva, 21007 Huelva, Spain; angel.sanchez@dfaie.uhu.es

**Keywords:** strip detectors, interstrip effects, interstrip energy resolution, GRIT detector

## Abstract

In this work, we present a novel approach for improving the energy resolution from particles impinging on the interstrip regions of silicon strip detectors. We employed three double-sided strip detectors from the GRIT array and a triple α-source under laboratory conditions. The results showed that the interstrip resolution depends not only on the impinging side but also on whether it is a P- or an N-interstrip. We obtained the interstrip energy resolution down to 0.4%, and, depending on the scenario, the resolution was enhanced by a factor of 2. We believe that this new rotation method allows for the possibility of applying particle identification methods on interstrip events, which in most cases are dismissed during data recording.

## 1. Introduction

Silicon strip detectors (SSDs) are basically an evolution of simple pad detectors driven by the need to localize a particle in space, i.e., spatial resolution. The matrix coordinate system is achieved (in the case of a double-sided SSD, DSSSD) by orthogonally dividing the side electrodes into micrometer strips with their corresponding implants underneath. The strips are separated by an insulator material, usually SiO_2_, creating what is known as the interstrip region or just the interstrip. It is believed that one of the first SSDs was tested by Heijne et al. in 1980 [1]. The first SSDs had a number of thin strips which were connected to the readout electronics, but also some intermediate, unconnected strips between these readout strips. By using the pulse height information, the spatial resolution was improved thanks to the charge division due to the capacitive coupling between adjacent strips. The number of intermediate strips, the interstrip capacitance and the signal-to-noise ratio will limit the final spatial resolution [2,3]. Reducing the interstrip width below the projected size of the collected charge cloud will not help to pinpoint the particle impinging coordinates. Furthermore, the proximity of two strips will influence the collection process since the generated carriers (i.e., electrons and holes) may be subjected to uneven depletion depths or variable collecting fields. Effects such as opposite polarity pulses have been reported and are attributed to charge injection from adjacent electrodes [4]. The interstrip capacitance should be larger than the bulk capacitance so that charge losses are avoided in the process of charge division. This is usually achieved with a small pitch geometry, i.e., a <100 μm readout pitch [5]. Studies on how the proximity of strips affects the generated signal are still ongoing. Researchers are employing not only small-particle α-sources but also particle beams of different natures, which in some cases are of micro-beam size and vary from light particles like protons to heavier ones such as lithium and oxygen. Experiments were performed addressing the total energy recovery of interstrip events [6] and the effective interstrip width [7], revealing that depending on the injection side and the bias voltage, these events could be recovered by summing the signals arriving in parallel from two adjacent strips. It has also been shown that interstrip effects, if not taken care of, diminish the capability of particle identification at energies below 5 MeV, especially on the injection side of the detector [8]. Meanwhile, as reported in [9], digitized interstrip signals have demonstrated the possible detrapping of trapped carriers over times on the order of microseconds. The high segmentation of DSSSDs has been used to discriminate β particles from charged particle events, as explained in [10], where after an implantation process, a detector was employed for counting the number of implanted nuclei and detecting the α+d branch. For these purposes, interstrip signals were utilized to identify double-hit events. Alternatively, lasers have been employed as radiation sources for interstrip studies, used for generating minimum ionizing particles. For example, a recent work that utilized a femtosecond laser beam has revealed an “unexpectedly strong signal” induced near p-stop implants that depends on the laser power [11]. On the whole, SSDs have nowadays become common devices in many large detector systems (e.g., ALICE [12], ATLAS [13], BlueSTEAI [14], GRIT [15]) with the aim to study, for example, the shell structure, the shape evolution, the scattering and reactions in inverse kinematics, etc.

Although the interstrip regions of silicon detectors have been the subject of much research work, none of these works have addressed the energy resolution that can be obtained from interstrip events, i.e., particles impinging in this region. This was our motivation. We believe that our work may impact particle identification techniques, which utilize the energy information as one of the distinguishing parameters. It may also impact the way experimental data are managed. The knowledge gained could also be applied to quality control processes for silicon detector manufacturers. As we have found, the energy resolution of interstrips depends on the detector injection, and in some cases, it may well match the resolution of standard strips.

This paper is organized as follows: Section 2 describes both the experimental framework and the method employed to estimate the energy resolution of the interstrips. Section 3 then combines the results and evaluates the interstrip energy resolution in different scenarios. Finally, Section 4 presents our conclusions.

## 2. Materials and Methods

In this section, we first describe the detectors employed in this work and how the data were acquired. Then, we explain how the energy shared by two adjacent strips was “translated” so we could evaluate the energy resolution from the interstrip. This work was carried out on the detector GRIT test bench, which is fully described in [16]. For the sake of clarity, we will briefly describe the setup. Figure 1 shows one of the detectors in the chamber facing the α-source; the readout cables can also be seen. The detectors were placed in a vacuum chamber at $10^{-5}$ mbar. Their signals were taken out of the chamber and fed into the preamplifiers for final digitalization and data processing.

### 2.1. Silicon Strip Detectors

Three DSSSD units were put under test. These are part of the GRIT system [15] and they were manufactured by Micron Semiconductor Ltd. (Lacing BN15 8SJ, UK) [17]. The detectors are of the same dimensions (rectangular) and of equal characteristics (neutron transmutation doped silicon, 500 μm thickness), as can be seen in Table 1. Although the strip separation is the same on both the ohmic and junction sides (60 μm), the strip widths are different, at 715 μm for the P-strips (junction side) and 765 μm for the N-strips (ohmic side), i.e., a 6.5% less strip area on the junction side. Another difference between the junction and the ohmic sides of the detector is the presence of the two p^+^ implants between the ohmic strips or N-strips, the purpose of which is to prevent a “short circuit” between adjacent n^+^ implants.

### 2.2. Particle Impinging Scenarios

When any DSSSD is exposed to radiation, the impinging particles will indifferently enter through a strip (i.e., an aluminium layer on top of its corresponding implant) or through the passive SiO_2_ layer between two adjacent strips, i.e., the interstrip. This in turn will generate a number of electrical signals depending on these two cases. Figure 2’s left panel shows a schematic representation of a small area of the detector, where the front (vertical) and the rear (horizontal) strips are labelled; the impinging scenarios are also labelled from Ⓐ to Ⓓ. A close-up of the detector showing vertical strips (whiter stripes) can be seen in Figure 2’s right panel. In scenario Ⓐ, the particle is impinging on one of the front strips and the generated charges will be gathered by not only this strip but also by its orthogonal rear one. Therefore, two electrical signals are generated. In scenario Ⓑ, the particle is impinging on one of the front strips and the generated charges will be gathered by this strip and the two neighbouring rear orthogonal strips. This is due to the splitting of charges between the two rear strips since the particle trajectory lies within the rear interstrip, and therefore, the carriers are attracted by the two electric fields. Hence, three electrical signals are generated. In the case of scenario Ⓒ, the particle enters through a front interstrip and the generated charges will be shared by the two front strips and also by the two rear strips; again, the particle trajectory also lies within the rear interstrip. Four electrical signals are generated in this scenario. Finally, scenario Ⓓ presents the case where the particle impinges on the front interstrip but the particle trajectory lies within a rear strip, and therefore, two electrical signals are generated by the front strips and one by the rear strip; i.e., a total of three electrical signals are generated. We have summarized these scenarios, the number of electrical signals created and the dead layers that particles must punch through in Table 2. This section helps the reader to understand the energy resolution obtained from these different scenarios which we will explain in the result section.

### 2.3. Selection of the Interstrip Events

A selection of “true” interstrip events prior to the data analysis was performed. Despite having the four digitizers synchronized (64 electronic channels each), the information was stored in an asynchronous way to reduce the dead time. The energy value from each generated electrical signal was recorded alongside the engaged electronic channel number (from 1 to 64), its digitizer board number (from 1 to 4) and a timestamp. By managing these four arguments, we could select the interstrip events of any given strip pair. The procedure was as follows. After the acquisition, the events were sorted in ascending order according to their timestamp. Then, we calculated the time differences between an entry ($i_{x}$) and each of the next three ones ($i_{x+1,2,3}$), e.g., $i_{x}$-$i_{x+1}$, $i_{x}$-$i_{x+2}$ and $i_{x}$-$i_{x+3}$; the produced histogram is shown in Figure 3’s left panel. We remind the reader that an impinging particle may generate up to four electrical signals or entries (Table 2). Next, we selected the entries of two adjacent strips within the same digitizer board; an example of this correlation is shown in Figure 3’s middle panel. It can be observed that, apart from the three distinct diagonal distributions, there are also other loci, especially in the top right corner. Eventually, by imposing a time difference (time window) between entries, we were able to select the interstrip events (Figure 3, right panel). To see all the entries that form the diagonals, the time window should be about 100 ns. By shortening this time window or just shifting it to lower or upper values, only some entries of the diagonals will be selected. Although the selection procedure removed the entries above the diagonal lines, which corresponded mostly to particles impinging on any of the two adjacent strips, there still are some entries below the diagonals that persist and that we believe are caused by the electronic chain. This selection process is unique to our acquisition setup and it should be adapted not only to the running setup but also to the kind of study being carried out.

### 2.4. The Rotational Interstrip Energy Resolution

As explained above, the charge sharing phenomenon takes place when a particle impinges on an interstrip. The ionization cloud created in the bulk is subjected to the electric fields of the two nearest strips and therefore part of these charges will go to one strip and the rest will go to the other. In other words, a single particle generates two electrical signals. In terms of energy, we can say that the total incident energy of the particle is divided between the two adjacent strips (electrodes) in a proportion that depends on the distance between the entrance point and the strip. If we were to plot the energies captured by one strip vs. the energies of its neighbour, assuming that all these interstrip events are produced by the same particles at a given energy, we will see a linear distribution of points. Figure 4’s left panel shows an example of the interstrip events recorded between strip 60 and 61, produced by particles emanating from a triple *α*-source, i.e., ^239^Pu + ^241^Am + ^244^Cm. Each of the diagonal distribution lines corresponds to one of the source’s particles, and as can be seen, they are parallel to each other, with a negative slope of 45°. If we were to obtain the energy distribution of these events, we would require their projection onto one of the axes, which will not yield any useful information. However, by rotating these distributions so they become perpendicular to an axis (Figure 4 middle panel), we could obtain a clear energy spectrum. The rotation of the distributions around a given center can be easily achieved by employing the following formulas:
(1a)$$x_{1}=\left[\left(x_{0}-x_{c}\right)\;cos\left(\theta \right)\right]-\left[\left(y_{0}-y_{c}\right)\;sin\left(\theta \right)\right]+x_{c}$$
(1b)$$y_{1}=\left[\left(x_{0}-x_{c}\right)\;sin\left(\theta \right)\right]+\left[\left(y_{0}-y_{c}\right)\;cos\left(\theta \right)\right]+y_{c}$$
where $\left(x_{0},y_{0}\right)$ is the point to be rotated, $\left(x_{c},y_{c}\right)$ is the center of rotation, θ is the rotation angle and $\left(x_{1},y_{1}\right)$ is the new point after rotation. Applying this to Figure 4, $x_{0}$ is the energy of strip 60 and $y_{0}$ is the energy of strip 61. We selected energies at the end of the diagonals ($x_{c}=5486$; $y_{c}=0$) and after the projection of the rotated data, a second calibration was performed to align the peaks with their energies (Figure 4, right panel). This rotation method should be applied to interstrip events that produce the charge sharing distributions in the form of straight lines or diagonals. This scenario can be found, for example, in direct kinematics experiments (e.g., $E_{lab}=20$ MeV with a resolution below 1%, a ^12^C beam, a ^208^Pb target and a beam spot size on the target of a few millimetres), where the DSSSD is placed at a laboratory angle of about 20° and less than 50 cm away from the target. Under these conditions, particles will hit the interstrips at a high rate (since we are dealing with elastic scattering, a high cross-section at forward angles) and the energy dispersion of the elastically scattered beam impinging on the interstrips will be very low (because the angular spread of the interstrips is very small). Therefore, it is very likely that there will be narrow diagonals when plotting two neighbouring strip energies.

## 3. Results and Discussion

The results shown in this section correspond to one of the three detectors tested. Nevertheless, they are a representative sample; i.e., they accurately represent the characteristics of the other two detectors. We start this section showing the energy resolution of each single strip so we have a reference when comparing it with the resolution of the interstrips. Then, we present the interstrip energy resolutions for different injection scenarios.

### 3.1. Strip Energy Resolution

Figure 5’s left panel shows an example of the triple α-source spectrum obtained from a single strip. The three main peaks of ^239^Pu, ^241^Am and ^244^Cm are accompanied by smaller “satellite” peaks, which is a good indicator of the energy resolution of the detector. By Gaussian fitting the ^241^Am peak, we obtained the energy resolution of the 128 P-strips (blue circle) and the 128 N-strips (red square) for both ohmic (right panel) and junction injections (middle panel). Table 3 shows some of the detector’s working parameters, such as the depletion voltages ($V_{depletion}$), the applied bias voltages ($V_{bias}$) and the total leakage currents at bias ($I_{total}$). It also shows the average full width at half maximum (FWHM) of the ^241^Am peak for the 128 P- and 128 N-strips. The obtained average energy resolutions (FWHM) for both the P- and N-strips are quite similar, and their differences are not significant. For the sake of clarity, we have not included the FWHM uncertainties in the table. We can say that the overall energy resolutions are within the acceptable range for particle identification techniques [18,19].

### 3.2. Interstrip Energy Resolution

Figure 6 shows the rotational interstrip energy resolution obtained from both junction and ohmic injection, more precisely scenarios Ⓒ and Ⓓ depicted in Figure 2. We remind the reader that we applied the rotation procedure explained in Section 2.4 in order to obtain the spectra. The ^241^Am peaks have been Gaussian-fitted (red dashed), and both the mean and the sigma (σ) of the fitting are shown in the top left corner of each panel. From these fittings, we obtained the FWHM =2.35σ and the energy resolution in percentage, i.e., the FWHM divided by the peak centroid. The best resolution was obtained for the P-interstrip (i.e., the interstrip between two adjacent P-strips) in the ohmic injection, with a FWHM =23.6 keV or 0.43% (top left panel). However, the worst resolution was obtained in the same scenario for the N-interstrip, with FWHM =57.1 keV or 1% (bottom left panel). For junction injection, an FWHM =36.2 keV or 0.66% was obtained for the P-interstrip (top right panel) and an FWHM =32.3 keV or 0.59% was obtained for the N-interstrip (bottom right panel). At first glance, we observe interstrip energy resolution values close to those of the strips (see Table 3), and therefore, it may be inferred that we can obtain valuable information from the interstrips. Next, we explain what is limiting the interstrip resolution in each case.

Figure 7 shows two different interstrip scenarios, one for junction injection (left panel) and another for ohmic injection (right panel). The SiO_2_ layer on the junction side uniformly covers the entire width of the interstrip, while on the ohmic side, SiO_2_ only intermittently covers $3/5^{ths}$ of the interstrip width. We also point out that, under the interstrip on the ohmic side, there are p^+^ implants (p-stops), which are non-existent on the junction side. Having noticed this, we clearly see that particles impinging on interstrips will undergo different energy losses depending on whether they enter into the bulk material through the SiO_2_ layer or not. We employed LISE++ code [20] to calculate the energy loss and straggling of an α-particle (^4^He) with an energy of 5.486 MeV passing through different materials. The results of these calculations are shown in Table 4. The main differences come from the energy losses and the lateral spread, with higher losses when the particle crosses the 1 μm SiO_2_ layer. This is translated into a shorter penetration depth in the bulk material of the detector, i.e., ≈0.8 μm less than not going through this layer. In other words, the energy deposited by particles impinging on N-interstrips (Figure 7, right panel) will have larger fluctuations than that of particles impinging on P-interstrips (Figure 7, left panel) due to the non-uniformity of the SiO_2_ layer. The final energy distribution (spectrum) in this case will have two contributions: the one from particles passing through the SiO_2_ layer ($\sigma_{1}$) and another one from those entering directly into the Si material ($\sigma_{2}$). Therefore, the N-interstrip energy FWHM fluctuation will be $\sigma_{f_{N}}=(\frac{3}{5}\sigma_{1}^{2}+\frac{2}{5}\sigma_{2}^{2}{)}^{1/2}$, while the P-interstrip fluctuation will be $\sigma_{f_{P}}=\sigma_{1}$. This is evident when comparing the sigmas obtained for the junction injection P-interstrip ($\sigma_{f_{P}}=15.4$ keV; Figure 6, top right panel) and for the ohmic injection N-interstrip ($\sigma_{f_{N}}=24.28$ keV; Figure 6, bottom left panel). We can conclude this part of our study by saying that the interstrip resolution depends on the injection side, with the worst resolution when the particles are impinging on the N-interstrips; this is due to the non-uniformity of the SiO_2_ layer. The energy resolutions obtained from interstrips opposite the impinging side are always better than their counterparts (Figure 6, top left and bottom right panels). This is because particles with trajectories that lie within these interstrips mostly enter through the opposite strip, i.e., through the Al electrode, and only a small number enter through the SiO_2_.

### 3.3. Resolution Comparison: Rotation vs. Addition

The total energy of interstrip events is usually recovered by adding the energies obtained from adjacent strips, i.e., $e_{total}\;=\;e_{\left(n\right)}\;+\;e_{\left(n+1\right)}$. The final recovered energy will be smaller than the incident particle’s energy due to the losses and the fact that the generated charges will be read by at least two different electronic channels. Therefore, in order to calculate the true incident energy, we need to take into account all these losses, i.e., $e_{true}$ = $e_{total}\;+\;e_{loss}$. This seems a straightforward method (addition) to calculate the incident particle’s energy. However, the energy fluctuation or sigma will limit our power of resolution and hence will cause us to deviate from the true mean energy. Figure 8 shows the spectra obtained by adding the energies of two neighbouring strips. From the Gaussian fitting parameters corresponding to the ^241^Am peak, we observed that there is an energy shift towards lower values for the four scenarios. This shift is due to energy losses and also due to the calibration procedure; in the next section, we present a method to counteract these. The junction injection scenarios (Figure 8, right panels) showed higher losses than the ohmic ones (Figure 8, left panels) due to the SiO_2_ layer, as we explained in the previous section. Table 5 summarizes the energy resolution values obtained from both the addition and rotation methods and also the enhancement factor. The column showing the resolution by adding the two energies presents two values: the first one (Cal^†^) corresponds to the values obtained employing the strip energy calibration, while the second (Cal^‡^) is obtained using the interstrip energy calibration explained in the next section. For all the scenarios, we obtained a better energy resolution when applying the rotation method regardless of the calibration procedure used. It is also noted that the enhancement factor is more dependent on the type of interstrip rather than the injection site. More precisely, the enhancement factors are always higher for N-interstrips no matter the injection side. This is expected since N-interstrips present the worst energy resolution, as explained above.

### 3.4. Particle Energy Determination within the Interstrip

As shown in the previous section, the total energy of an interstrip event cannot be recovered with sufficient accuracy by simply adding the energies of the adjacent strips. This is mainly due to the calibration procedure that uses the signals generated by particles impinging on a strip (i.e., Al + Si) and not on an interstrip (i.e., SiO_2_ + Si). The energy values of the sources employed for the calibration purpose, such as a triple α-source, and/or the energies of the beam given by the accelerator can be used to calibrate the ADC channels processing the signals generated by interstrip events. As discussed earlier, the sharing of charges between two adjacent strips produces a linear correlation in the form of diagonal distributions with a negative slope of −1 (we are assuming that there are no phenomena involved that prevent the charges from reaching the electrodes). This implies that we can use the middle of these diagonals (i.e., equivalent to half the energy of the source) to calibrate the ADC channels reading these events. Figure 9 shows how this should be achieved. The line y=x cuts the α-diagonals at their central energy values. These crossing coordinates are obtained by Gaussian fitting the projection of all the diagonal’s points on the y=x line (inset in Figure 9). In this way, we ensure that, once the calibration is complete, the addition of the energies of both channels yields the total energy of the impinging particle. The energy resolution values obtained after applying this calibration (Cal^‡^) are shown in Table 5. Although there is a slight improvement in resolution compared to the strip calibration (Cal^†^), the resolution from the rotation method is greater. As previously mentioned, to calibrate the detectors in a reaction experiment, apart from the standard triple α-sources, one could use, for example, the elastic scattering of the incident beam at different energies to obtain new diagonals and therefore extend the energy dynamic range in the calibration.

## 4. Conclusions and Future Works

In this work, we have presented a novel method of obtaining the energy resolution of particles impinging on the interstrip regions of silicon strip detectors. This new rotation method may improve the energy resolution by up to a factor of two when compared to the traditional method of adding energies. We have also discovered that the worsening of the interstrip resolution for particles impinging on the N-interstrips is due to the uneven coverage of the SiO_2_ layer. We believe that this new approach of improving the interstrip energy resolution will help to develop new techniques for particle identification that use energy information as a distinctive parameter. As for future work, new proposals both for analyzing the existing data from previous experiments and for dedicated, new, in-beam experiments are being planned. 

## Figures and Tables

**Figure 1 sensors-24-02622-f001:**
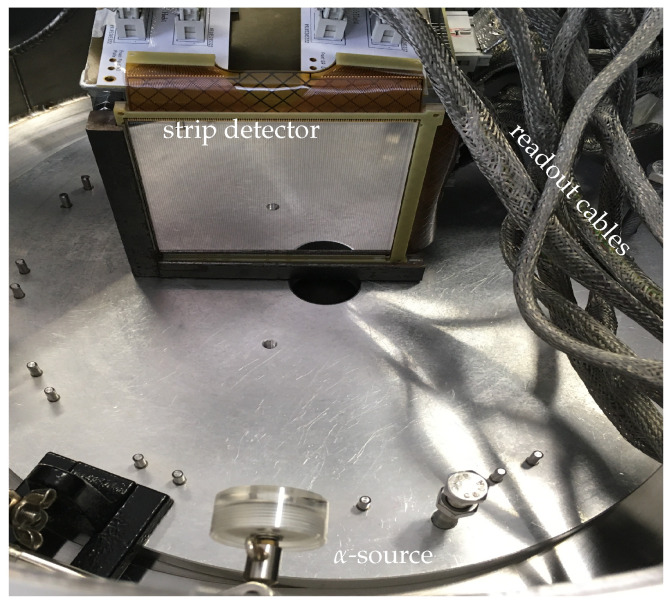
Picture of the vacuum chamber’s interior. The detector is facing the α-source. The signals are taken out of the chamber by the readout cables to the electronic chain.

**Figure 2 sensors-24-02622-f002:**
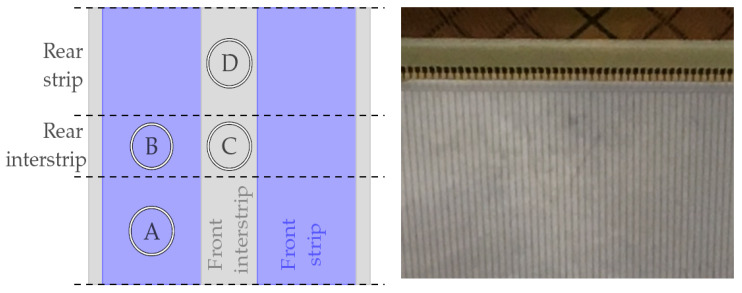
Strip detector structure layout. **Left panel**, schematic of two adjacent strips and its interstrip with four different marked areas A–D. **Right panel**, a close-up of the strip detector employed: vertical strips and interstrips (darker).

**Figure 3 sensors-24-02622-f003:**
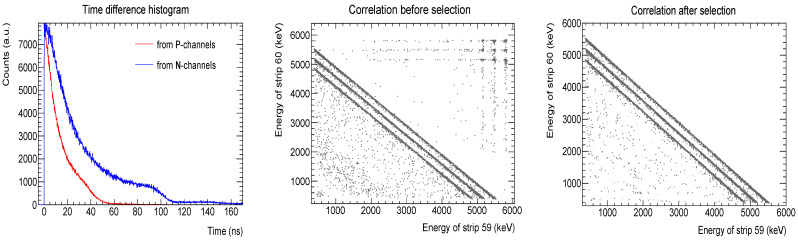
Selection of the interstrip events. **Left panel**, time difference histogram between an entry and the next three ones. Energy correlation between two adjacent strip before (**middle panel**) and after (**right panel**) the selection procedure.

**Figure 4 sensors-24-02622-f004:**
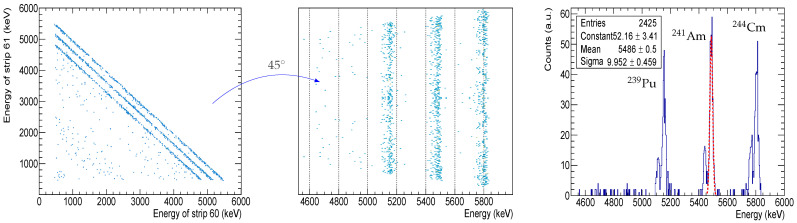
The rotational interstrip energy resolution. **Left panel**, the energy plot of strip 60 vs. 61. The data set is rotated by 45°, as shown in the **middle panel**. **Right panel**, calibrated projection of the rotated data set, i.e., an energy spectrum. The ^241^Am peak fitting parameters are shown in the stat box.

**Figure 5 sensors-24-02622-f005:**
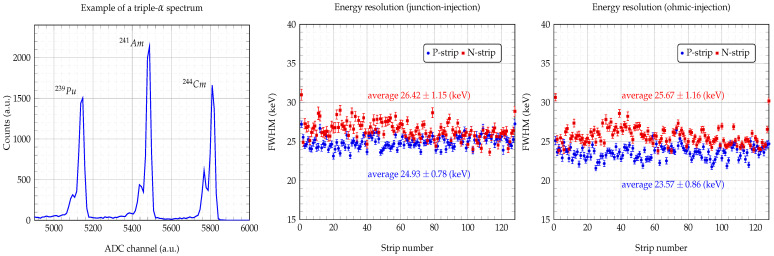
Energy resolution per strip. Example of a triple-α spectrum from a single strip (**left panel**). ^241^Am full width at half maximum (FWHM) of each P- and N-strip for junction (**central panel**) and ohmic injection (**right panel**) scenarios at a bias voltage of 140 V.

**Figure 6 sensors-24-02622-f006:**
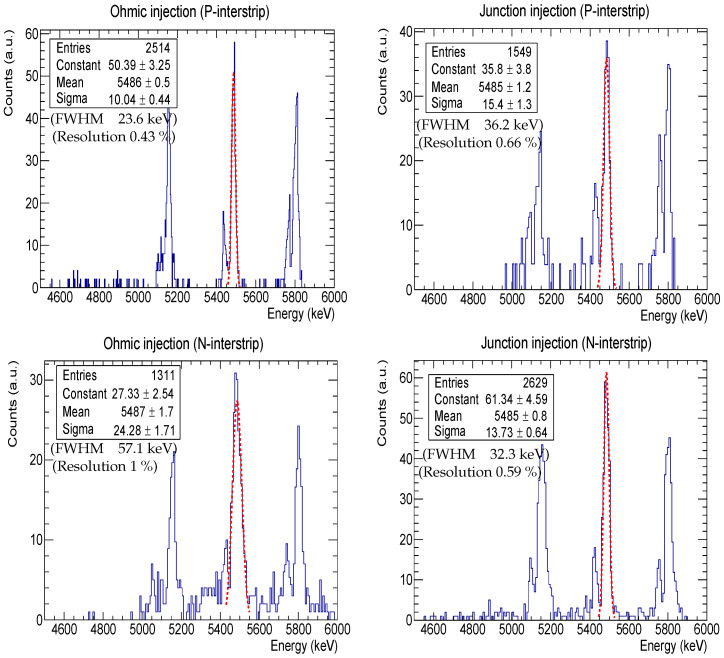
Interstrip energy resolution for both junction (**right panels**) and ohmic (**left panels**) injection scenarios. The Gaussian fits for the ^241^Am peak are shown as a red dashed line; their mean and sigma values are also indicated in the statistic box.

**Figure 7 sensors-24-02622-f007:**
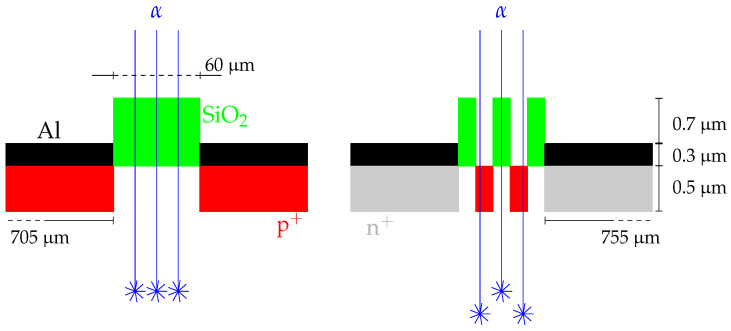
Interstrip scenarios (not to scale) for alpha particles impinging on the junction (**left panel**) and on the ohmic (**right panel**) side. The ohmic side has two insulating p^+^ implants.

**Figure 8 sensors-24-02622-f008:**
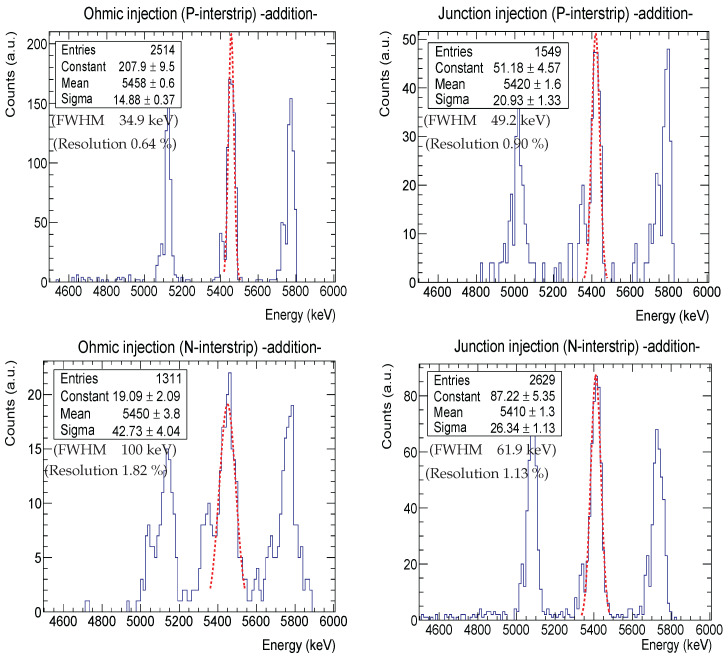
Interstrip energy resolution obtained by adding energies for both junction (**right panels**) and ohmic (**left panels**) injection scenarios. The Gaussian fits for the ^241^Am peak are shown as a red dashed line; their mean and sigma values are also indicated in the statistic box.

**Figure 9 sensors-24-02622-f009:**
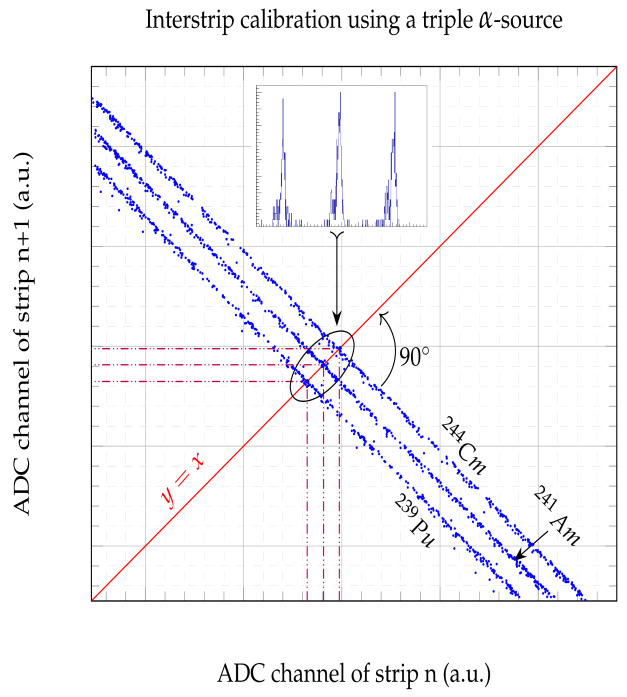
Interstrip energy calibration using a triple α-source. The line y=x cuts the α-diagonals at their central energy values, e.g., at 5486/2=2743 keV for ^241^Am. The crossing coordinates are obtained by Gaussian fitting the data projected on the line (inset). These crossing points correlate the ADC channels with the energies of the α-particles.

**Table 1 sensors-24-02622-t001:** Main detector characteristics as given by Micron Semiconductor Ltd. [17].

Parameter	Description
Wafer type	NTD silicon 〈100〉, size $6^{\prime \prime }$, No.: 3539-14, -17 & -19
Substrate type	N-Type Silicon (NTD 5-degree off axis)
Resistivity	7–10 kΩ·cm
Thickness	500µm, total thickness variation of ±1µm
Detector type	Double-sided ion implanted totally depleted structure
Implantation	Boron for junction side (P-strips) and phosphorus for ohmic side (N-strips)
Strip no.	128 P-strips and 128 N-strips, i.e., 128×128
Strip pitch	765µm for P-strips and 815µm for N-strips
Strip separation	60µm for both P- and N-strips
Isolation	2 P-stop structure between N-strips
Metallizing	Aluminum 3000 Å
Dead layer	<1µm
PCB	105.5×108.6 mm^2^
Outputs	via Molex 53916-0808 (Mouser Electronics, Mansfield, TX, USA) embedded in Kapton flex cable

**Table 2 sensors-24-02622-t002:** Summary of the particle impinging scenarios as shown in Figure 2.

Scenario	Electrical Signals Generated	Entrance Dead Layer
Ⓐ	1 front + 1 rear	Al (0.3µm) + implant (0.5µm)
Ⓑ	1 front + 2 rear	Al (0.3µm) + implant (0.5µm)
Ⓒ	2 front + 2 rear	SiO_2_ (1µm)
Ⓓ	2 front + 1 rear	SiO_2_ (1µm)

**Table 3 sensors-24-02622-t003:** Detector parameters. Average FWHM for the junction and ohmic injection scenarios.

Detector	V*_depletion_*	V*_bias_*	I*_total_*	FWHM*_junction_* P-/N-Strip	FWHM*_ohmic_* P-/N-Strip
3539-14	115 V	140 V	510 nA	24.4/25.3 keV	23.3/24.2 keV
3539-17	120 V	140 V	540 nA	24.9/27.6 keV	23.5/26.1 keV
3539-19	115 V	140 V	650 nA	24.3/26.2 keV	23.6/25.7 keV

**Table 4 sensors-24-02622-t004:** LISE++ simulation results for ^4^He at 5.486 MeV.

Material	Thickness	Energy Loss	(σ) Energy Straggling	(σ) Angular Straggling	Lateral Spread
SiO_2_ ^††^	1 µm	112.8 keV	1.4 keV/u	4.61 mrad	0.0031µm
Si implant ^†^	0.5 µm	69.1 keV	1.6 keV/u	4.22 mrad	0.0014µm
Al ^§^	0.3µm	46.9 keV	1.3 keV/u	3.35 mrad	0.00068µm

^††^ density 1.724 g/cm^3^. ^†^ density 2.321 g/cm^3^. ^§^ density 2.702 g/cm^3^.

**Table 5 sensors-24-02622-t005:** Energy resolution comparison: addition vs. rotation.

Scenario	Resolution (%) by Adding Cal ^†^/Cal ^‡^	Resolution (%) by Rotating	Enhancement Factor ^§^
Ohmic injection (P-interstrip)	0.64/0.55	0.43	1.5/1.3
Ohmic injection (N-interstrip)	1.82/1.74	1	1.8/1.7
Junction injection (P-interstrip)	0.90/0.82	0.66	1.4/1.2
Junction injection (N-interstrip)	1.13/1.03	0.59	1.9/1.7

^†^ Strip calibration. ^‡^ Interstrip calibration. ^§^ Rounded to one decimal place.

## Data Availability

Data are contained within the article.

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
