# Peer review of "Energy Resolution from a Silicon Detector’s Interstrip Regions"

_sensors, 2024, doi:10.3390/s24082622_

Round 1

Reviewer 1 Report

Comments and Suggestions for Authors

m-minor, M-major comment

m line 31 - on going->ongoing

m 98 - the scenario C - scenario D

m 170 detector i.e., - detector, i.e.

Author Response

Dear reviewer, thanks for your comments. All the typos have been corrected.

Reviewer 2 Report

Comments and Suggestions for Authors

The manuscript discusses a new method to recover the so-called interstrip event in Single and Double Sided Silicon Strip Detector (SSD & DSSSD) and improve energy resolution of the detector for such events. The method is described in some detail and is applied to data obtained with  a standard triple alpha source.

Despite the somewhat limited tests and the lack of data with a beam, I find that this study is of relevance to the reader of this journal as the topic is important for the community that uses such a kind of detectors.
However, before the paper can be accepted for publication, I would like the authors discuss some of the issues that arose while reading the manuscript.

Major remarks
1) About rotational method.
As stated in section 2.3, the rotational method needs the determination of the coordinates of the rotation center (xc, yc), and, if I understood correctly, each energy requires a different rotation center. The method works fine when a triple alpha-source is used. In this case you know a priori that all the particles that hit the detector have just three energies. Therefore you can easily find three loci corresponding to the three energies of the alpha-particles (i.e the diagonal bands) and for each of them is possible to define a centre of rotation.
However I wonder how the authors intend to apply this method in an experiment. In a generic esperiment the detected particles have a wide range of possible energies. When you find an interstrip event (an event where you have two electrical signals from two adiacent strips), how will you define the corresponding center of rotation (xc, yc)?
In other words, data from an experiment will not be divided in few clear loci
like in Fig. 3 where you have three diagonal bands. But they will be spread all over the x-y quadrant. Therefore you cannot apply  the method you used in the paper that is: "For xc and yc we selected energies at the x-end of one of the diagonals". Because will not be any diagonal. 

Which procedure you think to use to find a rotational centre for each interstrip event?

2) about the energy determination.
In Section 3.2, Fig. 5 shows the energy spectra of the three alpha peaks for interstrip events in scenarios C and D when particle enter the detector from both sides after the rotation method is applied. The energy of the three peak is smaller than the correct energy.

For example, let consider fig. 5 Ohmic injection (P-interstrip) picture. The energy of the 241Am peak is 3.108 MeV while it shoud be 5.486 MeV. The energy missing, more than 2 MeV, is not compatible with the energy
losses in the dead layers tha the authors calculated with LISE that cannot account for more that 200 keV. I think that the problem lies in the definition of the coordinates of the rotation center: x_c, y_c.

Even in the best case "Junction injection (P-interstrip)" the energy of the peak corresponding to the 241Am is at 5.112 MeV far from the correct one more the 350 keV.  The same peak is found at 5.422 MeV using the addition method, just 64 keV.

The resolution is obviously important for the application you cited as particle identification. Anyway, also the determination of the energy is equally important, if the method fails to get the correct energy I think it can hardly be used for particle identification.
You should improve your method in such a way it can provide also the correct energy.

Would be very usefull if the authors could provide the figure obtained using the addition method for the other cases, that is:
Ohmic injection (P-interstrip)
Ohmic injection (N-interstrip)
Junction injection (N-interstrip)
Is the mean energy of the peak recostructed in a better way than with the rotation method?

Minor remarks

1) Text must be careful checked for typo, I just list some of them

line 49: can obtained -> can be obtained
line 62: was -> were
line 72: demissions -> dimensions
line 98: in this line scenario C shoud replaced with scenario D
line 117: ration -> rotation
line 146; Mean and Sigma should not be written in capital
line 152: there is a break of the line that should not be there.
line 182: oposite -> opposite

2) In the abstract and in the following you speak about the particle identification methods. Could you be more precise? There are several methods used to idetify particles in some case the energy resolution is very important in some it is not.

3) From Fig. 3 it is possible to infer that the energy of strip 61 ranges between 2400 to 5800 keV while in strip 60 the energy ranges between 500 and 3000 keV.
Why there is this difference? Do you have data also for other strips that can clarify this?

4) line 3: It is not clear what do you mean with "under laboratory conditions".

5) lines 119, 120: "For x c and y c we selected energies at the x-end of one of the diagonals, for the sake of clarity." Since the value of the parameters x_c and y_c is critical for the rotational method, I think could be important to describe in some detatils how do you choose these values and, possibly,
if and how the results are sensible to this choice.

6) Just a suggestion. In fig. 5 could be usefull to write FWHM instead of sigma in the plot.  This would make easier, for the reader, to compare the value with those of table 3 and fig. 4 where the FWHM is used.

7) Would be very useful, if possible, to quantify the percentage of type A, B, C and D. In other words how many of the total number of events are interstrip? Are they 1% or 20%?

8) The Ohmic injection (N-interstrip) picture (bottom left corner) of Fig.5 do not looks like a spectrum of a triple alpha source as the one in Fig.4 or the others in Fig.5, could you explain why?

9) Are you able to distinguish, or have you tought to a way to distingush between an interstrip event and a real coincidence (I mean two particles that hit two neighbouring strips)? This could be important for experiment where you have an high rate on the detector or for those experiment where you
look for physical coincidences of particles that have a small realative momentum.

Author Response

Dear reviewer, thanks for your time and comments. We believe that your valuable feedback has greatly improved our new manuscript. Please, find attached our reply to your questions.  

Reviewer 3 Report

Comments and Suggestions for Authors

I believe that the work cannot be accepted for publication, since the proposed technique, aimed at improving the energy resolution of interstrip events, seems to be affected by an incorrect procedure. In fact, when a distribution is rotated, so as to become perpendicular to the energy axis referring to a strip, the energy scale is maintained unchanged and not normalized as should be done (see fig. 3). In fact, the centroids of the energy distributions corresponding to the alphas is the same of that acquired by the strip, that depends by the charge sharing among the two adjacent strips. Fig. 5 shows four energy spectra acquired in different experimental conditions, and each one reports different energy values of the centroids, wherever the spectra require an appropriate calibration which centers the peaks at the correct energy values, thus allowing the correct evaluation of the widths. Furthermore, the authors compare the resolutions using the widths in keV and not in percentage, as should be done in these cases. For example in fig. 5 states that the best resolution is in the top left pad, but the 0.7% FWHM resolution is comparable to the 0.8% in the top right pad. In summary, the rotation procedure needs to be reviewed with correct assignment of the energy read from both stripes, in order to correctly evaluate the effectiveness of this technique. Personally don't see why simply rotating the axis should reduce the energy resolution, so I'm skeptical about the effectiveness of this technique. But I invite the authors to thoroughly review the article and adequately reanalyze the data, in order to demonstrate incontrovertibly whether the technique works as declared.

Furthermore, the document needs to be deeply revised in other several parts. 

Just to give you an idea, I report a list of observations that however is far from exhaustive.

- Introduction, page 2. There are several topics apparently unrelated to each other and to the main topic of paper. The motivations must be well described, harmonized with each other and with the main topic.

-Fig.3. As already mentioned, the energy scale of the right pad plot depends on how the charge is shared in the interstrip region. A suitable recalibration should be required.

- 133. The reference to scenarios A and B could cause confusion. In fact, in fig.4 the resolution for junction and ohmic injection in single strips is shown (scenario B is for interstrip on the back), while scenarios A and B refer only to the injection on the front side.

- 173-179. The main cause of the fluctuation in the N interstrip spectra is due to the different energy losses between the n. 3 layers of SiO2 and the n.2 p+ regions, and to a lesser extent by the sigma of both energy distributions. Furthermore, in sigmafN the contribution of sigma1 and sigma2 should be adequately weighted (3/5 and 2/5).

- 201-202. The enhancement factor needs to be re-evaluated by comparing the correct energy resolutions.

- 220. A future work should also report the improvement in terms of particle identification. If it were to be demonstrated that after rotation the resolution improves (I have many doubts about its effectiveness), this argument alone cannot justify an article, but it must be completed with the clear demonstration of improvements in particle identification.

Author Response

(The authors gave the same response as above.)

Round 2

Reviewer 2 Report

Comments and Suggestions for Authors

Dear authors

In this manuscript you present a novel method aimed to recover the events generate by ions impinging in the so-called "interstrip region" of a DSSSD with particular focus on the energy resolution of such class of events.
You apply this method on a data set taken with an alpha-source finding that you are able to reconstruct the energy of the alpha particle with a
resolution better than the "addition method".

I believe that the method you present has good potentiality, but they are not fully explored.
In my opinion, at this stage, the method is not fully mature and lack of interest for the community because
1) cannot be applied to a general case where you have a continous energy spectrum
2) did not prove, in a convincing way, the capability to determine the energy of the particle with a sufficient accuracy.

You did not address in a convincing way to these two issues:
To the first one, you postpone the answer to new studies to be done in the future.
To the second one, you just applied a second energy calibration that actually does not solve the problem.

Both the questions are relevant for the aim of the method. If you want to identify particles you need to be able to use the method in a real experiment
where the particles have a continuos spectrum of energies (not just three energies as with an alpha-source). You need also to know with a good accuracy the particle energy otherwise you fail in the identification of the particles.

I think that, if the above-mentioned issues are addressed in convincing way, the work can be improved in a significant way, it could rise a good interest
in the community, and it could be reconsidered for the pubblication, but, unfortunately, I cannot suggest the paper for the pubblication in the present form.

Author Response

Dear reviewer 2, once again thank you for your comments and suggestions. We have know included a whole new section 3.4. Particle energy determination within the interstrip, where we explain how to calibrate the system to accurately obtain the energy of the particles. We also explain how this could be extended to a real experiment. We sincerely hope that with this new section we have proven, in a convincing way, the capability to determine the energy of the particle.

We respectfully disagree with the statement “cannot be applied to a general case …”. By selecting the interstrip events that produce the charge sharing distributions (diagonals), as explained in section 2.3, we can then applied algorithms to identify particles within any given energy range.

We strongly believe, that without this work, there will not be any experiment addressing the particle identification within interstrips. We have taken the first steps towards that goal, and therefore, we believe that this paper is of interest to the scientific community dealing with silicon strip detectors.    

Reviewer 3 Report

Comments and Suggestions for Authors

The paper has been modified as requested. It can be published as is.

Author Response

Thank you for your comments, we believe our work has greatly benefited from them.